# Active Power Control of Retrofit LED Tube Lamps for Achieving Entitled Energy Savings in View of the EU Ban on Mercury

**Shounak Roy** [1] **and Michael Krames** [1,2,*]

1 Seaborough Research BV, 1098 XG Amsterdam, The Netherlands
2 Arkesso LLC, Palo Alto, CA 94306, USA
* Correspondence: m.krames@seaborough.com

**Abstract:** The performance of commercially available retrofit LED tubes intended for the replacement of linear fluorescent lamps is measured and analyzed with respect to "real-world" installed electronic ballasts, as ascertained through field recovery from installed bases, such as office buildings, parking garages, and industrial installations in western Europe from 2018 to 2020. Results show a wide variation in lamp power draw, which not only thwarts the energy-saving and climate protection aspects of the LED retrofit solution but also poses potential safety risks. Given the EU's goals under the Restriction of Hazardous Substances (RoHS) directive to phase out mercury-containing fluorescent lighting, starting from September 2023, the situation is alarming. We show that this lamp power spread is due to the fundamental differences in impedance between fluorescent lamps and LEDs, in combination with the passive nature of the driver electronics that are currently employed in commercially available LED tube lamps. In response to this disparity, a novel driver topology including active power control (APC) is introduced, which shows that the power-spread problem can be avoided, and we offer a manufacturable solution. A prototype retrofit LED tube lamp incorporating this APC driver technology is shown to deliver safe and predictable energy savings, outlining a path toward guaranteeing the expected return-on-investment and positive environmental impact of the solid-state lighting replacement of mercury-containing linear fluorescent lamps.

**Keywords:** lighting; LEDs; linear fluorescent lamps; retrofit lamps; solid-state lighting; LED drivers; electronic ballasts; mercury lamps; mercury ban; RoHS

## 1. Introduction

### 1.1. Background

Lighting consumes approximately 17% of the total source electricity consumption in commercial buildings worldwide [1]. Until now, linear fluorescent lighting has been the lighting solution of choice in commercial and industrial settings and offers the largest technical potential in terms of energy savings via conversion to LED products, which are continually improving [2,3]. However, fluorescent lamps continue to be considered reasonably efficient, have a decent working lifetime, and have manageable maintenance costs. The most widely used linear fluorescent lamps (LFLs) for commercial applications are the T8 and T5 fluorescent tubes. There are currently approximately 210 million fluorescent tubes sold across Europe each year, consuming around 1.8 TWh [4]. Most lighting in US commercial buildings is provided by T8 and T12 LFL troffers (ceiling fixtures), with an installed base of 1.1 billion units [2]. In the EU, there is an installed base of more than 2.2 billion such units [5], about 40% of which are configured with electronic ballasts, while in the US, this fraction is more than 90% [6]. For the EU and the US, taken together, there is an installed base of approximately 1.9 billion LFL luminaires configured with electronic ballasts.

*1.2. Environmental Policy and the "Mercury Ban"*

Over the past decade, LED lighting has become the primary choice for replacing traditional light sources, mainly due to the promise of substantial energy savings and relatively long lifetimes, coupled with a bright outlook from other aspects, such as human-centric lighting [7]. This has prompted the conversion of traditional light sources to LEDs, a mercury-free global transformation in the spirit of the Minamata Convention on Mercury [8] and backed by government incentives and policies worldwide [9,10]. For example, the EU's RoHS directive on the phase-out of mercury-containing fluorescent lighting (part of the so-called "mercury ban") is slated to become enforceable in September 2023. This ban will present a significant climate-protective aspect, saving an additional 20 TWh/year, and will avoid the presence of five metric tonnes of toxic mercury [11]. It will also trigger a massive conversion in the market from fluorescent lamps to their LED counterparts [12].

*1.3. LED Replacement: Luminaire vs. Lamps*

Based on energy savings return-on-investment payback and the undeniable benefit of combating climate change, building owners today are already compelled to convert their lighting installations to solid state, but a complete exchange from fluorescent lighting luminaires to integrated LED luminaires is not an easy path to traverse, mainly due to:

1.　High initial costs of LED luminaires (compared to LED lamps [13]);
2.　High downtime for installations (especially when lighting for 24 h per day);
3.　High cost of labor (removing old luminaires and replacing them with new);
4.　Cannibalization of prior investments in fluorescent troffer electronic ballasts;
5.　Low fluorescent lamp replacement costs (the mercury ban is not yet enforced).

It appears obvious that a simpler way forward is to replace the linear fluorescent tubes with retrofit LED tube lamps, thereby realizing the much-awaited energy-saving benefits without the penalties mentioned above. This "plug and play" retrofit solution is the quickest path to the adoption and realization of energy savings, as it is the lowest-cost option and produces the least amount of electronic waste. Indeed, there is a plethora of retrofit LED tube products for electronic ballasts on the market (called "HF-only" or "Universal" tube lamps in the EU). However, as shown in this paper, we find they do not deliver the claimed energy savings, and their turn-on compatibility and performance vary heavily, depending on the legacy electronic ballasts already installed inside the fluorescent lamp luminaires that are available to house them.

## 2. Materials and Methods

*2.1. Field-Recovered Ballasts*

To investigate real-world retrofit conditions, rather than a carefully chosen set, the electronic ballasts selected for this work are ones recovered from office buildings, parking garages, and industrial installations in western Europe, where replacement of the linear fluorescent troffer luminaires had taken place. The ballasts were collected between 2018 and 2020 from electrical contractors and companies who had extracted them from these job sites. In total, more than 350 different T8 non-dimmable electronic ballasts that were meant for 1500 mm, 1200 mm, and 600 mm LFLs were recovered. Of these field-recovered ballasts, 129 were T8 58 W electronic ballast types meant for 1500 mm fluorescent lamps. A Pareto chart of these 129 ballasts, showing their manufacturers, is given in Figure 1. This large variety of electronic ballasts represents the actual diversity that can be expected in the field.

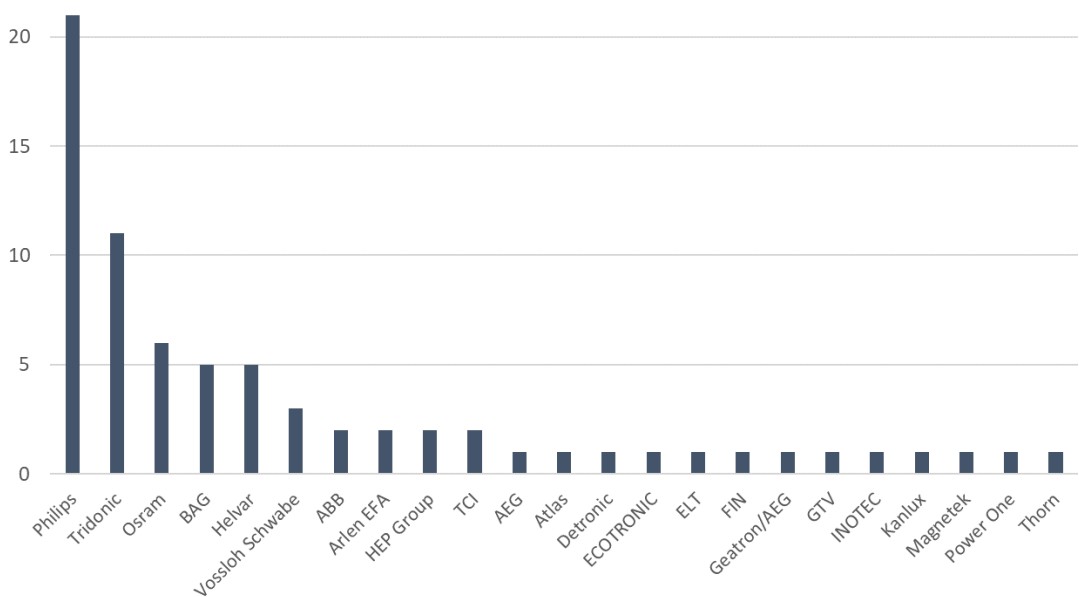

**Figure 1.** Pareto chart of the manufacturers for fluorescent tube luminaire non-dimmable electronic ballasts (*n* = 129) recovered from installations in western Europe between 2018 and 2020.

### 2.2. Lamp Test Apparatus

A test apparatus was designed and built at Seaborough Research BV, in Amsterdam, The Netherlands, to evaluate various retrofit LED tube products in combination with the field-recovered ballast types (*n* = 129). The schematic for this apparatus is shown in Figure 2, which also shows the measurement configuration. The apparatus accepts an electronic ballast, receiving its input from mains 230 Vac/120 Vac, which is connected to a linear tube lamp for characterizing the compatibility and performance.

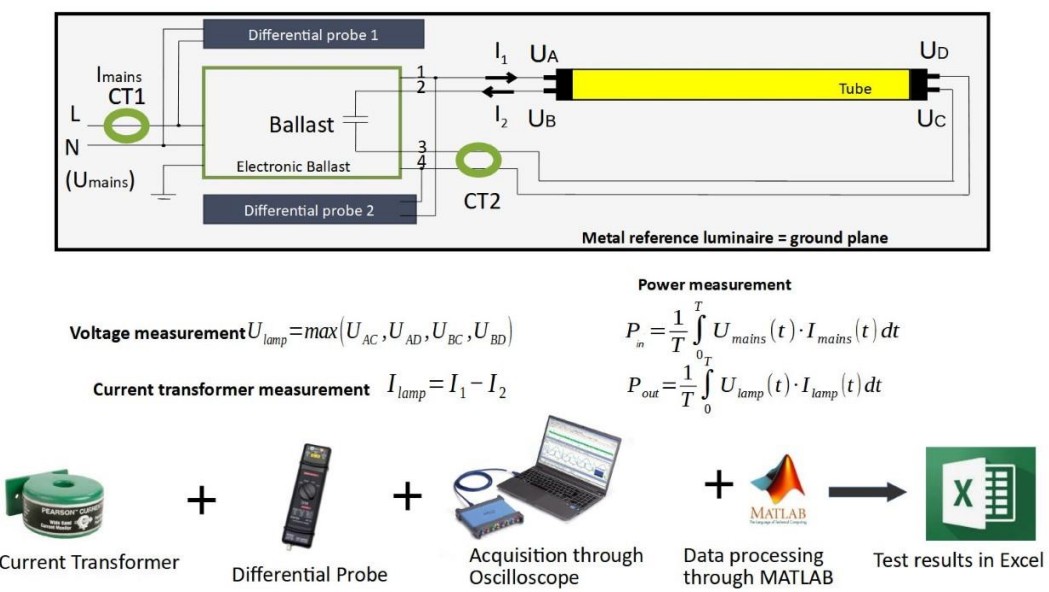

**Figure 2.** Configuration of the tube lamp–ballast combination test apparatus.

Since the output of the electronic ballasts carries high-frequency voltage and current, typically 40 Khz, the length, type, and thickness of the cables need to be the same when testing with different types of electronic ballasts. To prevent any difference in test results due to changes in the test setup, a fixed and fully automated test luminaire was built, and the lamp power was recorded in combination with each ballast type.

Figure 2 also shows the basic transducers used in the measurement of important parameters. The transducer marking and use are detailed below:

1. CT1 (Current Transformer 1) is used for ballast input current measurement;
2. CT2 (Current Transformer 2) is used for ballast output current measurement;
3. Differential probe 1 is used for input voltage measurement;
4. Differential probe 2 is used for output voltage measurement;
5. According to measurement standards, the highest voltage over the 4 outputs of an electronic ballast should be used. "Voltage swapping" between ballast outputs 1, 2, 3, and 4 is automated using a relay board and the measurement software;
6. An oscilloscope is used to receive the signals from the transducers;
7. The voltage and current values are collected and, thereafter, the MATLAB code is used to calculate the RMS power for both input and output measurements.

Figure 3 shows the actual test apparatus. The instrument list is shown in Table 1, while the measurement specifications are listed in Table 2.

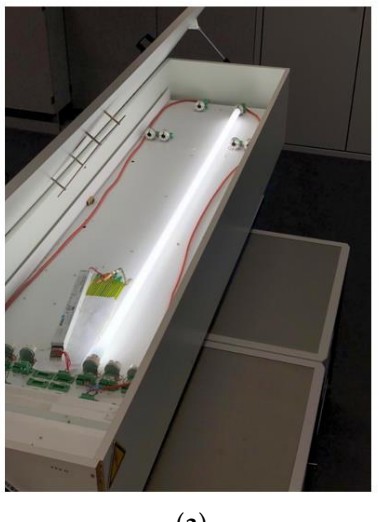
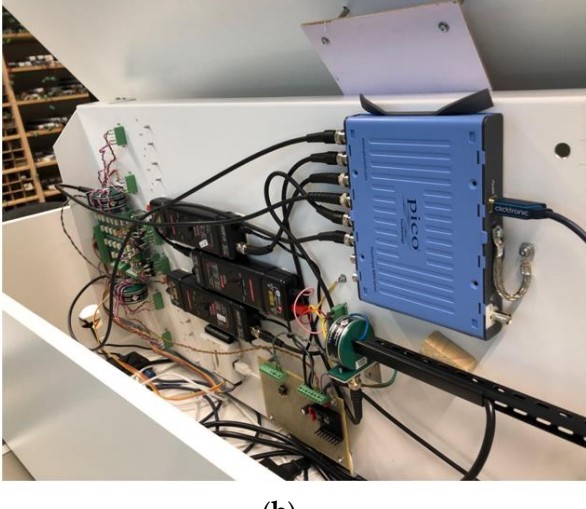

(**a**)　　　　　　　　　　　　　　　(**b**)

**Figure 3.** Actual tube lamp–ballast combination test apparatus at Seaborough Research BV in Amsterdam, NL: (**a**) top view showing a lamp/ballast combination in operation; (**b**) bottom view showing the included measurement instruments.

**Table 1.** Parts list for the LED tube/ballast test setup.

| Equipment | Manufacturer | Part No. | Function |
|---|---|---|---|
| Oscilloscope | Pico Technology Ltd. (St. Neots, UK) | 4824A | Capture input and output voltage and current signals from electronic ballasts. |
| Differential probe (3) | Pintek Electronics Co., Ltd. (Shulin City, Taiwan) | DP-25 | Measure ballast input and output voltage. |
| Current monitor (3) | Pearson Electronics, Inc. (Palo Alto, CA, USA) | 411 | Measure ballast input and output current. |
| AC power source | Keysight Technologies, Inc. (Santa Rosa, CA, USA) | AC6801A | Provide stable 50 Hz power to the ballast. |



**Table 2.** Test setup parameter measurement specifications.

| Parameter | Specification |
| --- | --- |
| Bandwidth | DC–20 MHz (−3 dB) Picoscope |
| voltage | DC–25 MHz (−3 dB) |
| current | 1 Hz–20 MHz (−3 dB) |
| Max sample rate | 80 MS/s |
| A/D resolution | 12-bit—(16-bit software-enhanced) |
| Buffer memory | 256 MS (shared between active channels) |
| DC accuracy | ±1% range ±300 μV Picoscope |
| Voltage | ±2% Diff. probe |
| Current | +1%/−0% Current transformer |
| 100 kHz accuracy | ±1% range ±300 μV Picoscope |
| Voltage | ±2% Diff. probe |
| Current | + 1%/−0% Current transformer |
| 1 MHz accuracy | ±1% range ±300 μV Picoscope |
| Voltage | ±2% Diff. probe |
| Current | +1%/−0% Current transformer |

For each test cycle, the LED tube and electronic ballast were both turned on and were run for 10 min to reach thermal equilibrium. Thereafter, the test cycle commenced, and the following parameters were measured:

- Ballast Input: U_rms, I_rms, P (system power), PF (power factor), frequency;
- Ballast Output/Lamp input: U_rms, I_rms, P (lamp power), frequency, PF (power factor);
- Thermal: ambient temperature.

The calibration of the measured parameters was performed with a PX8000 precision power meter (Yokogawa Test & Measurement Corp., Tokyo, Japan), capable of measuring power up to 1 MHz. The calibration test results showed all measured values within a ±2% accuracy.

We found the following considerations to be important during testing:

1. Localized voltage and current acquisition to minimize the common-mode rejection ratio (CMRR) and crosstalk;
2. Metal reference luminaire, to provide a ground connection for interference and leakage currents from ballasts;
3. Tightly wired luminaire, to enable accurate high-frequency measurements with leakage capacitances within electronic ballast specifications.

The equations used to calculate the voltage, current and power coming out of the electronic ballasts are shown in Figure 2. The lamp test apparatus and procedure can be used to evaluate both fluorescent and LED lamps. The test process and setup have been third-party-corroborated by DEKRA, and more details about the test apparatus and procedure can be found in the Supplementary Materials.

## 3. Results

### 3.1. Commercially Available Retrofit LED Tubes: Selection

Four popular, commercially available 1500 mm T8 retrofit LED tube lamps, sourced from three different manufacturers, were tested with each of the 129 field-recovered (non-dimmable) electronic ballast types using the test set-up described above. These lamps all have nominal wattages of 23–24 W and produce 2500–2800 lumens each, which is typical and meets the requirements for general office lighting. The specific model numbers and manufacturer's identities have been kept confidential.

### 3.2. Commercially Available Retrofit LED Tubes: Test Results

The resulting lamp-power measurements, in combination with the various ballasts, are summarized in Figure 4 (the upper four histogram charts). In Figure 4, the blue bars indicate the number of occurrences for a measured lamp power within a particular bin

(left *y*-axis), using single-watt bins that span from 0 to 56 watts (*x*-axes). The orange curves track the cumulative population of power bins (right *y*-axes). A lamp power reading of "0" was assigned to lamps that did not turn on when connected to a particular ballast.

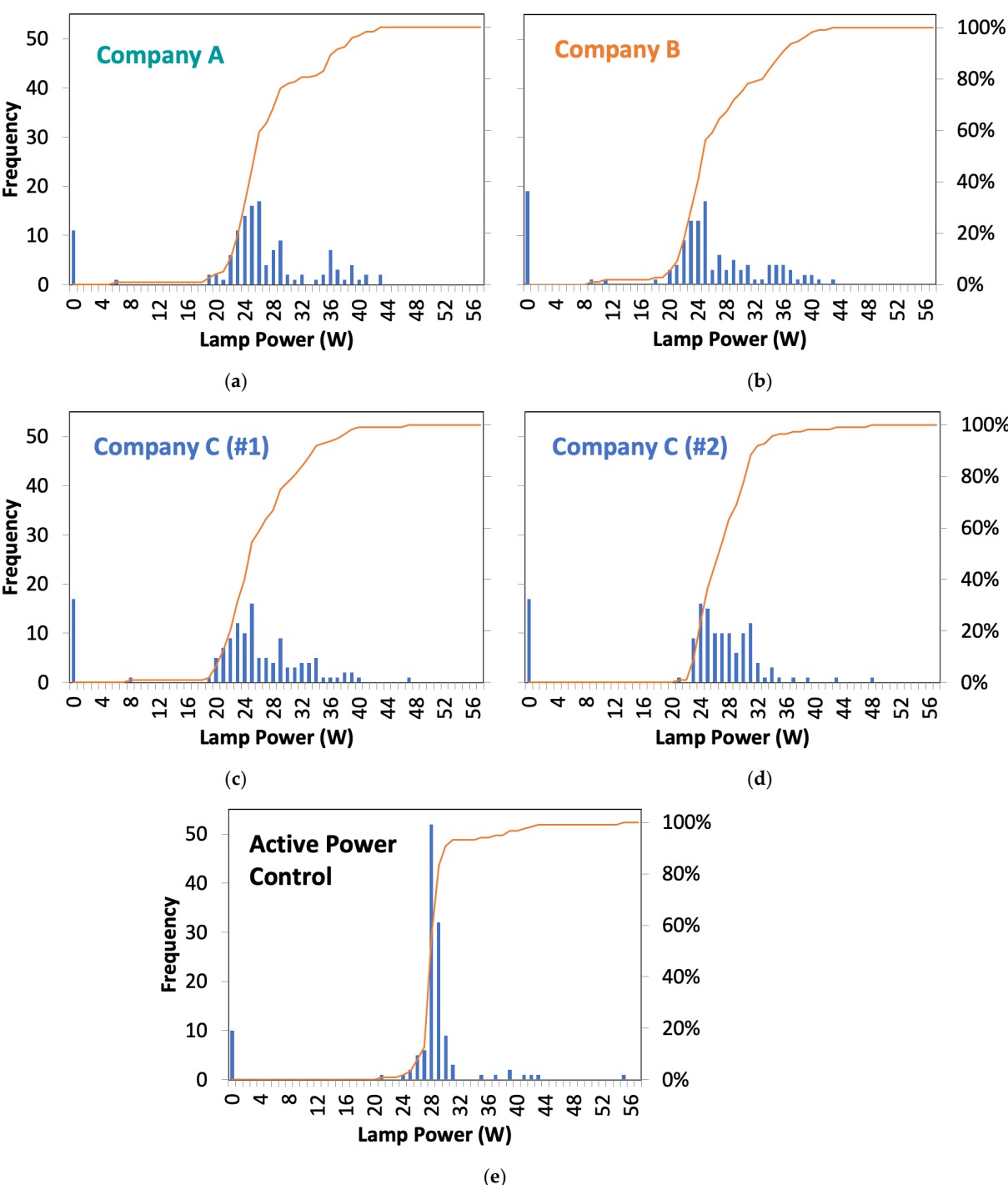

**Figure 4.** Histogram of lamp power measured in combination with all field-recovered ballasts (*n* = 129), for: (**a**–**d**) selected commercially available 1500 mm retrofit LED tube; (**e**) retrofit LED tube configured with an active power control (APC) driver, as described in this paper. The blue bars indicate histogram bin quantities (left *y*-axes), while the orange curve tracks the cumulative population (right *y*-axes).

The test results in Figure 4 show that retrofit LED tubes that are marketed to save approximately 50% of energy (vs. fluorescent lamps) can yield anywhere between 14 and

60% in terms of energy savings in the "real world", depending on the type of electronic ballast involved. This power spread has a significant impact upon actual vs. anticipated return-on-investment regarding energy savings and upon carbon footprint reductions that are expected to be achieved by conversion to solid-state lighting. Moreover, in the case of excessive power draw, safety concerns arise. Table 3 summarizes the LED tube lamp power spread across the *n* = 129 field-recovered ballast types.

**Table 3.** Summary characteristics of LED tube lamp power draw, in combination with the (*n* = 129) field-recovered electronic ballast types.

| LED Lamp Manufacturer | Declared Value (W) | Mean Power (W) | Stdev (W) | Stdev (%) | Fractionwithin +5/−10% of Declared Value |
|---|---|---|---|---|---|
| Company A | 24 | 27.1 | 6.0 | 22% | 42% |
| Company B | 24 | 26.6 | 6.1 | 23% | 44% |
| Company C (#1) | 23 | 26.2 | 5.5 | 21% | 29% |
| Company C (#2) | 24 | 27.2 | 4.2 | 17% | 37% |
| Active Power Control (APC)—this paper | n/a * | 28.6 | 3.8 | 13% | 82% |

* Since the prototype device does not have a "declared value", its power spread was calculated against its median power (27.9 W).

## 4. Discussion

From the test results, it is evident that the amount of energy saving depends completely on the type of electronic ballast that is present inside the luminaire and, for all the available commercial lamps that we evaluated, energy savings with electronic ballasts cannot be guaranteed. Indeed, for these lamps, more than half of the field-recovered ballast types resulted in a range of power consumption outside of the +5/−10% limits imposed by the new Ecodesign requirements, which went into effect in the EU in 2019 [14].

A derivative of the power spread problem is the "lumen spread" problem. The lumen output of the LED tube is directly proportional to the current drawn by the LED tube, which is proportional to the lamp power. (The assumption of the use of measured lamp power as a proxy for lamp output lumens was substantiated by a select set of lamp/ballast combinations, used in stabilized total flux measurements in a large integrating sphere, as detailed in the Supplementary Materials.) For lamps with a low power draw, this could present a perilous situation for installations where operational safety requires a minimum illuminance on the floor or other work surfaces. Correspondingly, an excessively large power draw could result in excessive glare, but more importantly, could lead to a reduced lifetime for the retrofit LED tube and, in the worst-case scenario, could present a thermal or fire hazard.

To illustrate the consequences of high-power spread, in Figure 5 we compare the total cost of ownership over a 7-year period for an installation having a 58 W T8 1500 mm fluorescent lamp, a 1500 mm T8 retrofit LED tube with 50% energy savings (when compared to a fluorescent lamp), and the same type of lamp with 14% energy savings (e.g., a lamp-ballast combination drawing high power, as is exhibited in certain cases, as shown in Figure 4). These calculations assume an electricity cost of EUR 0.15 per kWh (the latest non-consumer rate in the EU, including taxes [15]), a fluorescent lamp price of EUR 1.00, an LED lamp price of EUR 10.00, an operating time of 16 h per day, and a re-lamping cost of EUR 9.20 per lamp. (The details behind the total cost of ownership calculations can be found in the Supplementary Materials). The economical (and ecological) benefit of choosing an LED retrofit tube for energy saving is almost completely eroded in the case of an electronic ballast drawing high power, and the end-user has no way of assessing this before installation. Furthermore, we note that the situation in Figure 5 is the "best case" scenario regarding the high-power draw lamp (14% energy saving) since, for such a lamp, the efficacy is likely to be lower due to the high current draw (due to LED "droop" [16,17])

and elevated temperatures [18]. In addition, the lamp's lifetime can be expected to be shorter [19].

## Total Cost of Ownership Comparison

— Fluroescent Lamp 58W    — LED Tube (50% Energy Saving)    — LED Tube (14% Energy Saving)

**Figure 5.** Chart showing the potential disruption in the total cost of ownership if an LED installation involves electronic ballasts that draw much higher power, compared to the LED lamp's rated value.

### 4.1. Constant-Current Electronic Ballasts

Figure 6 shows a typical electronic ballast's internal circuit-block diagram, hereafter called constant-current electronic ballasts. The ballast has a power-factor correction (PFC) circuit block that ensures a stable 400–420 VDC power supply for the half-bridge switching circuit block, irrespective of changes to the mains line voltage, thereby resulting in an average constant current supply for the load. The load is typically a fluorescent lamp or a replacement LED tube. From the population of field-recovered electronic ballast, we see that ~80% of all electronic ballasts are constant current-type electronic ballast.

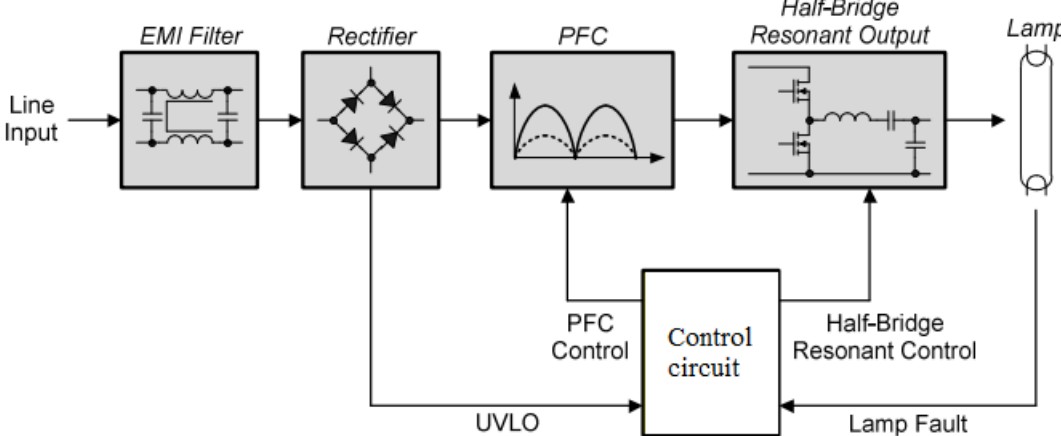

**Figure 6.** Typical circuit block diagram for a constant-current electronic ballast powering a fluorescent lamp.

Since most electronic ballasts are constant-current in nature, the power delivery is linearly dependent on the lamp voltage (Vlamp). For a typical 58 W T8 fluorescent lamp, the lamp voltage is around 104 V(rms) which yields around 50 W of lamp power and around

5000 raw output lumens. Since LEDs are more efficient compared to fluorescent lamp technology, typically, one needs 40–50% less power to achieve the same amount of useful lumens when in use. This is achieved by reducing the lamp voltage by about 50%, compared to the fluorescent lamp voltage. However, this is a huge step for the compensation and regulation loop inside the electronic ballast, as the system is not designed for such large load-voltage changes. This change in load voltage, in combination with a change in the load impedance from a fluorescent lamp (around 160 ohms) to an LED tube (around 3–5 ohms), causes a shift in the operating frequency of the electronic ballast. This also causes a shift in the delivered lamp current, moving either up or down. Since the LED lamp voltage is fixed, i.e., dependent on the number of LEDs, the same LED lamp in any given luminaire can draw significantly higher power with one brand, X, of installed electronic ballast compared to another brand, Y.

### 4.2. Constant-Power Electronic Ballasts

The second type of prevalent electronic ballast topology is a voltage-fed half-bridge quasi-resonant circuit, more famously known as self-oscillating topology, which is passive and has no lamp-current regulation technique. This causes the lamp current to depend heavily on the input voltage level and load impedance value. These ballasts are classified as "constant-power ballasts" because of their inherent nature of yielding similar power values, irrespective of the lamp voltage. From the population of field-recovered electronic ballasts, we see that ~20% of all electronic ballasts are constant current-type electronic ballasts. A typical ballast of this category is shown in Figure 7, which illustrates the schematic diagram of the half-bridge, self-oscillating topology [20].

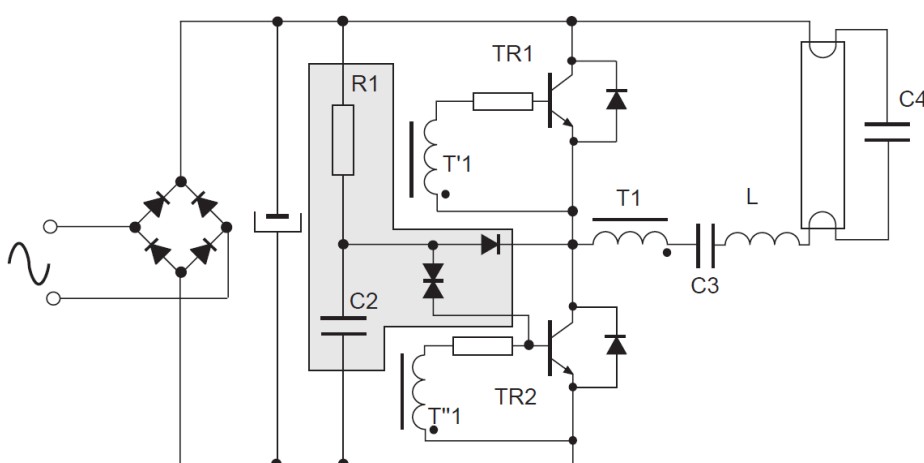

**Figure 7.** Typical circuit block diagram for a constant-power electronic ballast powering a fluorescent lamp.

For the constant-power circuit, the resonant frequency is:

$$Fsw = \frac{1}{2 \times \pi \times \sqrt{(L \times C_3)}}.$$

Apart from resonance, the impedance of the RLC series circuit is given by:

$$Z = \sqrt{[R^2 + (L\omega - \frac{1}{C_3\omega})^2]}$$

where R is the effective resistance of the load, i.e., the fluorescent lamp depicted in the figure.

At resonance, the $L\omega$ term equals the $1/C_3\omega$ and the two cancel each other out. Therefore, the impedance is minimum and equals the DC resistance, Z, where:

$$Z = \sum R$$

At resonance, the current in the circuit is at maximum and follows Ohm's law:

$$I = \frac{V_{CC}}{\sum}$$

where R is the resistance of the load, which was originally meant for a fluorescent lamp. Figure 8 shows a diagram of the equivalent circuit.

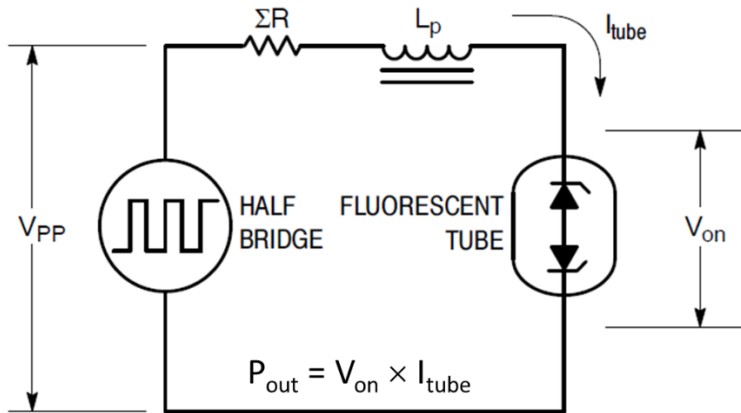

**Figure 8.** Equivalent circuit for a constant-power electronic ballast powering a fluorescent lamp.

After replacing the fluorescent lamp with an LED lamp, apart from the lamp voltage typically being 50% less than that of a fluorescent lamp, the impedance, Z, is around 3–5 ohms (due to LEDs acting in series and in a forward conduction mode) versus around 200 ohms for a fluorescent lamp (depending on the type of fluorescent lamp). There is a dramatic increase in the lamp output current level ($I_{tube}$, as per Figure 8), accompanied by a significant decrease in the lamp voltage ($V_{on}$, as per Figure 8), causing the output power of the ballast to hover close to the original fluorescent lamp power. The increase in the lamp output current is also accompanied by a change in operating frequency, which depends on the saturation flux density of the magnetics, T1, and storage time of the transistors, Tr1 and Tr2.

### 4.3. Active Power Control (APC)

As mentioned above, the current state of the art in terms of LED retrofit lamps for electronic ballasts does not employ any active power-correction mechanism that can react to the properties of the installed ballast.

We conducted an experiment to measure the change in the operating frequency of field-recovered electronic ballasts when the fluorescent lamp was replaced by a typical commercially available retrofit LED lamp. Figure 9 shows the change in frequency when the "load" is exchanged from a fluorescent lamp to an LED lamp having 50% less lamp voltage. The change in frequency causes the change in lamp current for LED tubes, leading to a large spread in lamp power consumption, depending on the electronic ballast.

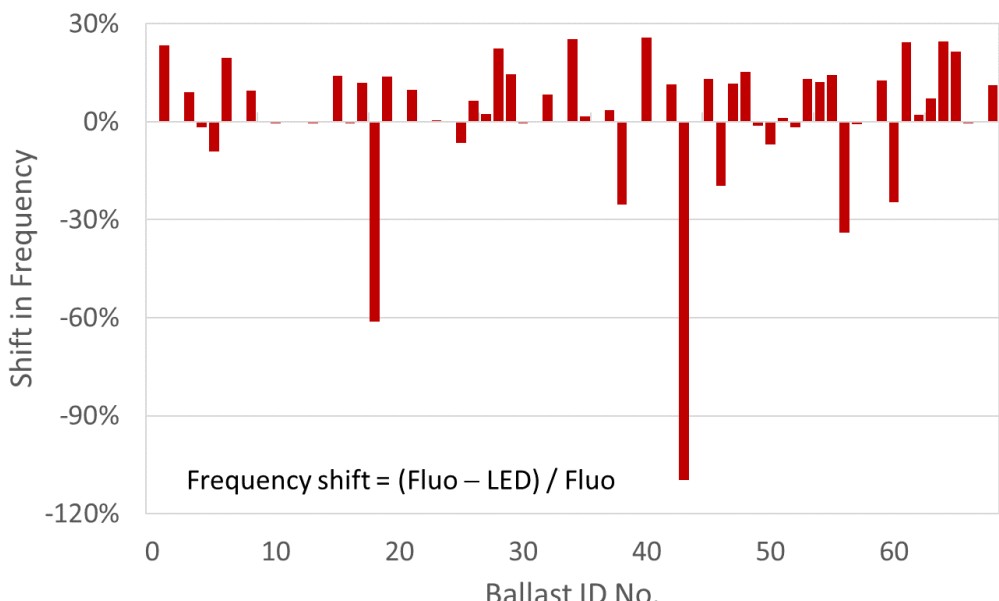

**Figure 9.** Resonant tank frequency shift between a fluorescent and an LED lamp for 1 × 58-watt ballasts.

For a typically chosen number of LEDs acting in series as the load, the LED tube power can easily vary between 20 W and 30 W. This is the fundamental problem faced by all types of retrofit LED tubes in the market. Since the LED load is fixed, depending on the make and brand of electronic ballast, the LED tube power cannot be guaranteed using a passive driver design, which is the case for all LED retrofit tubes on the market today.

To control the power draw from an electronic ballast, a novel idea is to provide a correctly adjusted LED lamp voltage that is proportional to the value of the lamp current per electronic ballast, thereby maintaining the same LED lamp power with all electronic ballasts. Figure 10 shows the change in LED tube power corresponding to a change in the number of LEDs in the series, for a sample of fifteen standard constant-current electronic ballasts. It is clear that the LED tube power is directly proportional to the number of LEDs acting in series.

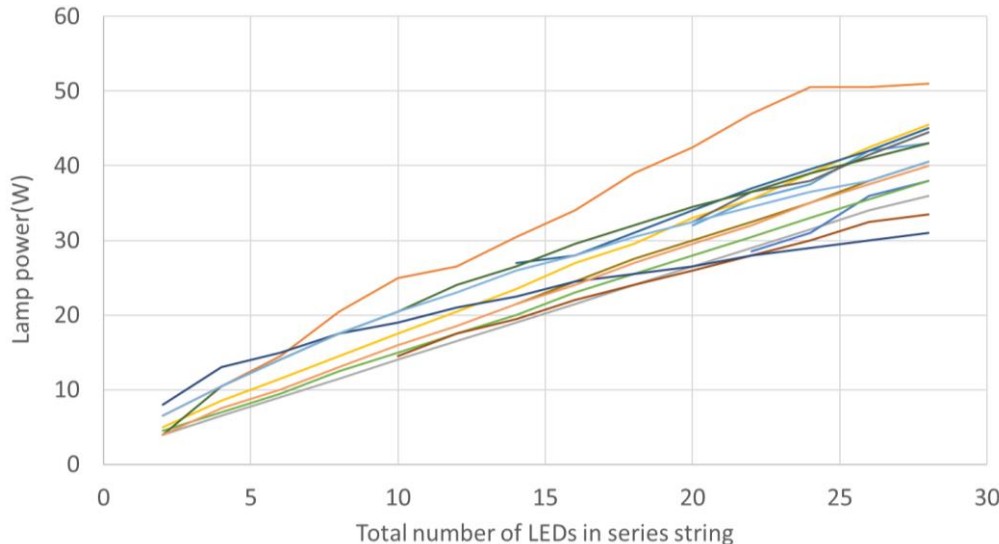

**Figure 10.** The LED lamp power draw from various electronic ballasts when the number of LEDs acting in series is varied.

Here, we introduce a novel active-power-control (APC) driver circuit; specifically, the idea includes:

1.  Determining the LED tube power needed to meet lumen and energy-saving requirements and choosing a total LED string size;
2.  The total LED string is split into two strings, string 1 and string 2, in an empirically determined ratio. This ratio is determined using the following steps:

    a.  LED string 2. The number of LEDs in string 2, when used as a load, should give an LED tube power equal to the rated LED tube lamp power with the highest output current electronic ballasts;
    b.  LED string 1. The total number of LEDs in string 1 and string 2, when used as a load together, should give an LED tube power equal to the rated LED tube lamp power with the lowest output-current electronic ballasts. (The string ratio is crucial and should ideally consider all electronic ballasts available for measurement).

3.  A semiconductor switch (transistor, MOSFET, or IGBT) is placed in parallel to string 1 (see Figure 11).

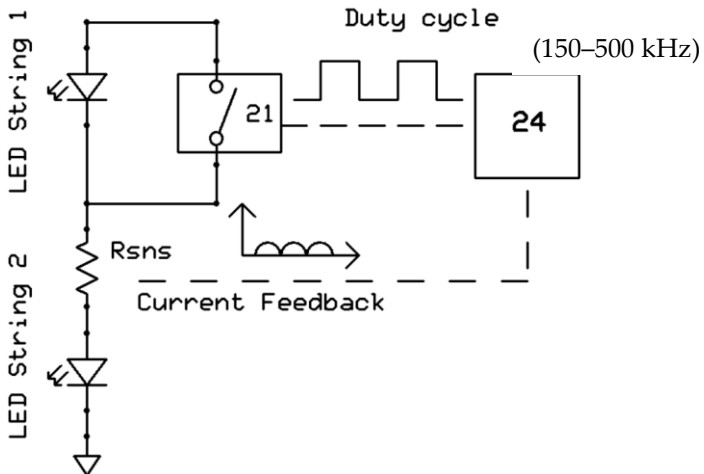

**Figure 11.** Simplified block diagram for the active power control (APC) driver circuit.

The new APC circuit works as follows: At t = 0, the power is switched on for the electronic ballast. A certain amount of current flows through the LED load. This amount of LED current is sensed by the sensing resistor, Rsns (Figure 11). This sensed current value and corresponding LED voltage are inputs fed into a microcontroller. The two sensed values, i.e., the LED current and LED voltage, are used to determine the actual LED power. The microcontroller has a reference value of power stored in it that is specific to the lamp type. The reference value is compared against the sensed value. Depending on the amount of difference between them, an error signal is generated, which is translated by the microcontroller in the form of a certain duty cycle. The microcontroller operates the semiconductor switch (marked as 21 in Figure 11), at a frequency much higher than the operating frequency of the electronic ballast, to avoid any triggering of fault conditions.

The duty cycle, determined by the microcontroller and based on the LED current, turns switch 21 on and off, effectively increasing or decreasing the total number of LEDs as the load. The duty cycle, therefore, determines the average voltage of the LED load over time. For a high-current electronic ballast, the LED current will be high, but the LED voltage (load voltage) will be low. For a low-current electronic ballast, the LED current will be low, but the LED voltage (load voltage) will be high. In both cases, the average LED power will be the same, thereby controlling the power actively, regardless of the specific electronic ballast type. Since the LED light output is directly proportional to power, this

APC solution also ensures that the same amount of light is emitted from the LED tube, regardless of ballast type.

For constant-power ballasts, a small addition involving an inductor and a switch is made to the power control circuit. The inductor is placed before the full-bridge rectifiers of the LED tube, thereby making it part of the resonant tank circuit, as described in Figure 8, which helps to add an additional impedance, XL (XL = $2 \times \pi \times f \times L$), to the total impedance, Z, thereby reducing the lamp current. The reduction in lamp current, along with switch 21 being fully closed (in most cases), creates the lowest voltage, causing the LED power to reduce significantly compared to that found in a situation without an inductor being present. Since the inductor is not needed for constant current electronic ballasts and because they lead to system efficiency losses and compatibility issues, a switch is placed in parallel to the inductor, enabling the microcontroller to turn the switch to "off" if a constant-power ballast is detected. The switch is normally set to "on", which keeps the inductor out of the circuit.

Using this method of controlling the power supplied by electronic ballasts, as described above, an APC driver circuit was designed, built, and incorporated into a commercially available test LED tube lamp. A 1500 mm universal prototype LED tube was made for the purposes of comparing the results. The lamp delivered ~3900 lm for an input power of ~29 W, a higher lumen output than that of the comparative commercial lamps simply due to the use of higher-efficacy LEDs. Testing of this prototype APC lamp with all 129 field-recovered ballast types was performed using the test apparatus described earlier in this paper. We note that the ON/OFF frequency of the APC of the LEDs occurs at >150 Khz, which does not exhibit any flicker perception to the human eye.

Figure 4e shows the power spread of the lamp incorporating the APC driver (bottom chart), for comparison with the commercially available tube lamps in Figure 4a–d. As summarized in Table 3, the results show a significant reduction in the power spread of the LED tube lamp when using the APC driver. In particular, the standard deviation is about half that of the commercially available lamps, while the fraction within +5/−10% of mean power (the Ecodesign requirement) reaches 82%, which is more than twice that of the standard commercially available LED linear tubes on the market today.

## 5. Conclusions

In summary, to explore real-world conditions for the LED retrofit replacement of linear fluorescent lamps, electronic ballasts (*n* = 129 different types) that were recovered from luminaires from various sites in western Europe were tested in combination with state-of-the-art commercially available LED tube products. We found that for all the commercially available products, lamp power varies dramatically, depending on the specific lamp/ballast combination. The range of lamp power draw can be as little as half, or as much as twice, that of the lamp's rated power or "declared value." In low-power situations, the installed lamp is not meeting the intended lighting design metrics. In the case of high-power draw, the lamp is delivering neither the energy savings nor the greenhouse gas emissions reduction that are expected by and promised to the end user, and this raises safety and lamp lifetime concerns.

To address this problem, we introduced an active-power-control (APC) driver design for LED retrofit linear tube lamps and evaluated this concept in a prototype. Unlike commercially available lamps, a prototype with an APC-driver circuit showed dramatically more uniform lamp power draw, with 82% of lamp/ballast combinations achieving the EU's 2019 Ecodesign guideline limits of +5/−10% for allowed power spread, a population fraction more than twice that of the commercially available retrofit LED tubes that we tested. This demonstration shows that, when properly designed, retrofit LED tube lamps are capable of delivering predictable power draw (and, thus, dependable energy savings) for all available sockets in the field, without the need to know or care about the legacy ballasts installed in various luminaires in different buildings over the decades. Such solutions are necessary to quickly and cost-effectively achieve the expected energy savings

and greenhouse gas emissions reductions promised by the transition from linear fluorescent lamps to mercury-free solid-state lighting, with its undeniable benefits.

## 6. Patents

Patent applications regarding the APC driver technology described in this paper have been filed, as listed in Table 4.

**Table 4.** Patent applications that were filed regarding APC driver technology, along with their status as of the date of submission of this manuscript.

| Type | Application No. | Filing Date | Status |
|---|---|---|---|
| PCT | WO2020084087A1 | 24 October 2018 | Published. In National Phase |
| Nationalized PCT (USA) | US20210385921A1 | 24 October 2019 | Publication Date: 9 December 2021 |
| Nationalized PCT (China) | CN112913328A | 24 October 2019 | Publication Date: 4 June 2021 |
| Nationalized PCT (Europe) | EP3871471A1 | 24 October 2019 | Publication Date: 1 September 2021 |

**Supplementary Materials:** The following supporting information can be downloaded at: https://www.mdpi.com/article/10.3390/su141610062/s1, Annex 1—Testing Process; Annex 2—Luminaire Lamp Holder Wiring Combinations; Annex 3—Total Cost of Ownership (TCO) Calculations; Annex 4—Lumen Measurements.

**Author Contributions:** Conceptualization, S.R. and M.K.; methodology, S.R.; analysis, S.R. and M.K; investigation, S.R.; writing—original draft preparation, S.R. and M.K.; writing—review and editing, S.R. and M.K.; supervision, S.R. All authors have read and agreed to the published version of the manuscript.

**Funding:** This research received no external funding.

**Institutional Review Board Statement:** Not applicable.

**Informed Consent Statement:** Not applicable.

**Data Availability Statement:** Not applicable.

**Acknowledgments:** The authors would like to thank Rob Klein and Jeffrey Wongsodikromo, who worked to create the control algorithms, samples, and test results.

**Conflicts of Interest:** The authors declare no conflict of interest.

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
