# Peer review of "Active Power Control of Retrofit LED Tube Lamps for Achieving Entitled Energy Savings in View of the EU Ban on Mercury"

_sustainability, doi:10.3390/su141610062_

Round 1
Reviewer 1 Report
After careful reading of the manuscript, I congratulate the authors for their research contribution and I suggest a few modifications for the betterment of the manuscript before accepting it for publication.
1. The title can be modified according to the novelty of the article.
2. The abstract should reflect the core novelty of the research and its significant contribution to the research community. Hence I request to modify it accordingly, which the abstract lacks now.
3. The introduction can be segregated into subsections with more references as per the research methodology opted for the current research. Also, the ned of the introduction should be clearly noted with "what research is existing, what are the gaps, what is current research novelty from the mentioned gaps".
4. The methodology should be represented in a schematic to improve the readability.
5. The results section should have a critical discussion (physics behind the results) on the obtained results, rather than just explaining the attained results.
6. The conclusions should be rewritten after including above all comments and they should be more clear and concise.
7. Spelling and grammatical mistakes are observed in the manuscript. It is suggested to have a careful English check before resubmitting the revised version.
Author Response
We thank the Review for their careful reading of our manuscript and for the very helpful feedback. We have taken the Reviewers' suggestions and incorporated them to improve our paper, as specifically noted below:
- We have changed the title and added "Active Power Control" to it.
- We have modified the abstract to reflect the background state of art in comparison to the introduced active power control approach.
- We have subdivided the Introduction to improve the narrative, and added more references regarding background information.
- We have replaced Fig. 2 with a higher resolution version to improve readability regarding the methodology.
- The Results section has been substantially revised and critical discussion about physics, etc., are put into the Discussion sections 4.1-4.4.
- We have edited the conclusion to make it more readable and clear.
- We have made several spelling/grammar corrections in the new revision.
Again, we thank the Reviewer for their helpful feedback and guidance towards improving the quality of our submission.
Reviewer 2 Report
In this work, 129 different types of electronic ballast (T8 58W) recovered from various installation sites in Western Europe were tested in combination with commercially available LED tube products. The authors found that, for all the commercially available products, lamp power varies dramatically depending on the specific lamp/ballast combination. To resolve this problem, and to deal with the energy savings and the green-house gas emissions reduction purposes, the authors have introduced an active-power-control (APC) driver design for LED retrofit linear tube lamps and tested this concept in a prototype. In my opinion, this paper is well written and it is suitable for publication in MDPI sustainability international journal. I have two concerns:
1. I suggest adding more references to support the main contribution of this work, practically, in the area of energy savings and the green-house gas emissions reduction.
2. Figure 2: Not clear, please make it clearer.
Author Response
We thank the Review for their careful reading of our manuscript and for the very helpful feedback. We have taken the Reviewers' suggestions and incorporated them to improve our paper, as specifically noted below:
- We have added significantly more references (for a total of 20) in the revised manuscript.
- We have replaced Fig. 2 with a higher resolution version to improve readability regarding the methodology
Again, we thank the Reviewer for their helpful feedback and guidance towards improving the quality of our submission.
Reviewer 3 Report
1. Please include more references to support your research gap.
2. Results are too brief and confusing for reader to understand the contribution of the APC with existing practice. Please articulate and add more results in the section.
3. Re-organise the Section 4 (4.1, 4.2 and 4.3) where the content would be good in Section 3.
Author Response
We thank the Review for their careful reading of our manuscript and for the very helpful feedback. We have taken the Reviewers' suggestions and incorporated them to improve our paper, as specifically noted below:
- We have added significantly more references (for a total of 20) in the revised manuscript.
- and 3. The results section has been revised to add content (including from Section 4 as suggested by the Reviewer) and to improve readability. The critical discussion about physics, etc., are in the Discussion sections 4.1-4.4 and have been revised for readability.
Again, we thank the Reviewer for their helpful feedback and guidance towards improving the quality of our submission.
Reviewer 4 Report
Authors present a technical solution, on which they have patent, for improving the LED retrofit solutions for conventional fluorescent luminaires using an Active Power Control - ACP. As there are still billions of ol fluorescent luminaires and if this produced the calculated amount of savings, this can be a very interesting solution.
The debate with pros and cons, regarding LED tube is already 5 years old, but this is the first time , when a solution is proposed. From the point of view of circular economy, it's a pity that a huge amount of conventional fluorescent luminaires become waste.
Here are my remarks:
- a scientific paper need to have at least 20 references
- any statement like 'Lighting consumes approximately 20%' Line 16 need to be referenced [x] mentioning from where you reached that conclusion
- some statement like 'High initial costs of LED luminaires' will raise controversy. This days you can find a LED luminaire with less than 15 Euro...
- assuption that 'electricity cost of €0,10 per kWh' it's history now. Have a look at https://ec.europa.eu/eurostat/statistics-explained/index.php?title=Electricity_price_statistics (you have a new reference)
- please rewrite more clear the conclusion
Author Response
We thank the Review for their careful reading of our manuscript and for the very helpful feedback, especially regarding the references. We have incorporated all the Reviewers suggestions, specifically:
- We have added more references and achieved the target suggested (20) by the Reviewer.
- We double-checked and updated this number and included a reference.
- Regarding relative costs, we are specifically referring to the high cost of luminaires vs. lamps. We have made the clarification in the text, and added a reference.
- We thank the Reviewer for bringing this to our attention, and have updated the electricity price to 0,15 per kWh (non-household rate including taxes) and have added the reference.
- We have rewritten the conclusion section to make it more clear.
Again, the thank the Reviewer for helping us improve the quality and impact of our submission.
Reviewer 5 Report
Dear authors
May I remind you that the primary purpose of a lamp is to produce light. Not to absorb power.
All the analysis performed on the electrical side is sound and safe and interesting. Nevertheless, NOTHING is analysed from the perspective of light luminous flux. Given the huge variability of LED conversion efficiency, the whole paper is void. The interactions between a ballast and its load are non-linear. Meaning nobody can reproduce nor explain your observations.
Basically, things go West at line 177. The LED luminous flux is mainly controlled by LED current, not LED power. The TCO you give is irrelevant, as it does not take into account the generated light level.
Fig. 7 use analog control. Since many years, the tendency is to have some kind of digital controller.
Fig. 11 requires more data. What is the typical range of switching frequency ? Did you take flicker into consideration ?
Author Response
We thank the Reviewer for their careful reading of our manuscript, and for helpful suggestions regarding improving it. We specifically address the Reviewer's points here:
- The lamps chosen for our study were commercial LED tube retrofit lamps for general office lighting. They have quite typical performance and are similar to each other regarding their specifications around the "declared" values (2500-2800 lm at 23-24 W). While the focus of our work was indeed on power measurements, in occasional flux measurements the authors observed that lamp light output was proportional to lamp power. This makes sense because increased power is mainly due to increased current, since LED voltage varies slightly with current. While we agree the interactions between the ballast and load are complicated, the implications are measurable (lamp drawn power) and we have described our test set-up and methodology in great detail so that one can repeat the results with their own lamps and ballasts. Furthermore (as noted in the paper), our test set-up has been inspected and validated by DEKRA.
- The TCO analysis is with respect to a lamp operating at its normal quiescent point. About that point, we generally see that lamp flux is proportional to lamp power, but agree with the Reviewer that the situation could be non-linear at the extremes of the distributions. That is, for very high power draw, lamp efficacy is likely to be lower due to high current draw and elevated temperatures. Also, lamp lifetime can be expected to be shorter. We have added sentences on this and thank the Reviewer for helping us improve the paper.
- We acknowledge since many years there is a tendency to have some kind of digital controller but around 15% of all field recovered ballasts are purely analog based. These ballasts are mainly part of the low cost market segment but evidently part of the installed base. Hence they cannot be ignored. We have added a reference for the diagram in Fig. 7.
- The authors recognize the pertinent question about flicker since the LEDs are being switches ON and OFF. However the ON / OFF frequency for APC of the LEDs are > 150Khz, which does not present any flicker perception to the human eye. Flicker percentages at 100Hz (twice mains frequency) were measured and found to be less than 5% for constant current electronic ballasts and less than 20% for constant power electronic ballasts. These values are not affected by the APC technology as the switching of the LEDs are happening at frequency from 150-500 kHz. We have commented on this in the MS and appreciate the Reviewer for helping improve our paper.
Again, we thank the Reviewer for their helpful and constructive comments which we have taken and used to improve the quality of our submission.
Round 2
Reviewer 1 Report
The manuscript can be accepted in its current form
Author Response
We are delighted the Reviewer believes the manuscript can be accepted in its current form. Again, we thank the Reviewer for their previous feedback which improved the paper.
Reviewer 4 Report
This version of the paper it's an improved one and can be accepted as it is
Author Response

(The authors gave the same response as above.)

Reviewer 5 Report
Where are effective lumen measurements ?
Author Response
Dear Reviewer,
Thank you for your question. Please find our answer in the attached file.
Kind Regards,
Mike Krames
